# Development of Aircraft Spoiler Demonstrators for Cost-Efficient Investigations of SHM Technologies under Quasi-Realistic Loading Conditions

**Markus Winklberger** [1,*], **Christoph Kralovec** [1] **and Martin Schagerl** [1,2]

1   Institute of Structural Lightweight Design, Johannes Kepler University Linz, Altenberger Str. 69, 4040 Linz, Austria; christoph.kralovec@jku.at (C.K.); martin.schagerl@jku.at (M.S.)
2   Christian Doppler Laboratory for Structural Strength Control of Lightweight Constructions, Johannes Kepler University Linz, Altenberger Str. 69, 4040 Linz, Austria
*   Correspondence: markus.winklberger@jku.at

**Abstract:** An idealized 1:2 scale demonstrator and a numerical parameter optimization algorithm are proposed to closely reproduce the deformation shape and, thus, spatial strain directions of a real aerodynamically loaded civil aircraft spoiler using only four concentrated loads. Cost-efficient experimental studies on demonstrators of increasing complexity are required to transfer knowledge from coupons to full-scale structures and to build up confidence in novel structural health monitoring (SHM) technologies. Especially for testing novel sensor systems that depend on or are affected by mechanical strains, e.g., strain-based SHM methods, it is essential that the considered lab-scale structures reflect the strain states of the real structure at operational loading conditions. Finite element simulations with detailed models were performed for static strength analysis and for comparison to experimental measurements. The simulated and measured deformations and spatial strain directions of the idealized demonstrator correlated well with the numerical results of the real aircraft spoiler. Thus, using the developed idealized demonstrator, strain-based SHM systems can be tested under conditions that reflect operational aerodynamic pressure loads, while the test effort and costs are significantly reduced. Furthermore, the presented loading optimization algorithm can be easily adapted to mimic other pressure loads in plate-like structures to reproduce specific structural conditions.

**Keywords:** aircraft spoiler; demonstrator; experimental strain analysis; quasi-realistic loading; SHM; composite sandwich; strain-based



## 1. Introduction

In aircraft engineering, scaled demonstrators are commonly used to represent parts, assemblies, or full-scale structures to allow cost-efficient testing. This is particularly true for aerodynamic development [1–3]. In addition, for many other purposes, small-scale models are used, e.g., aircraft design and flight testing demonstrators [4,5], demonstrators to develop flight-safety-critical systems [6], and technology demonstrators [7,8]. All of these demonstrators are mainly used to bridge the gap between experimental results of simple structures (e.g., airfoil test specimens, material testing specimens, test coupons for bonding tests), together with sophisticated models (numerical models applying the, e.g., finite element method (FEM)) for design calculations, and the final target structures in real applications [9]. Ideally, multiple experiments with such demonstrators in controllable test environments should then validate the preceding simulation results. In many cases, it is practical and cost-efficient to use multiple specimens or demonstrators of increasing complexity, e.g., first an airfoil, then a complete wing, and finally a full plane model, to validate models and conduct design studies. In structural aircraft design, a method known as the test pyramid has proven effective in addressing the rising costs associated

with the constantly increasing size and complexity in aviation industries [10–12]. The test pyramid for aircraft structures typically includes four layers: (I) coupons, (II) elements, (III) components, and (IV) full-scale airframe, beginning from the lowest and widest part (coupons; most generic structures; largest number of required tests) and ending at the tip of the pyramid (full-scale airframe; most complex structure; lowest number of required tests). A similar building block approach is the motivation for the present work. Hence, the layers of the test pyramid, including structural health monitoring (SHM), may have the following descriptions:

1. **Coupon and element level** (plate or beam-like structures)—development of sensors and SHM methods;
2. **Component level** (e.g., aircraft spoilers)—application of SHM to real structural components;
3. **Full-scale airframe level** (complete aircraft)—application of SHM systems to real aircraft structures.

In this context, an idealized demonstrator is located between the first and the second levels and should bridge the gap between simple plate- or beam-like structures and complex aircraft components (e.g., spoilers), which are expensive and laborious to acquire. Tailoring novel sensor systems and SHM evaluation methods reliably to structures of high complexity (e.g., a real aircraft spoiler) requires large and, thus, expensive test campaigns. A comparatively cheap idealized demonstrator together with an optimization methodology reflects an arbitrary behavior (and its associated effects, e.g., strain states and deformations are connected by the deformed shape of a beam or plate) of the full-scale component. Thus, a properly designed demonstrator structure and optimized loading allow one to imitate a specific behavior (and neglect all other effects) of parts of interest of complex structures and their loading in cost-efficient experiments. However, this also implies that the specific behavior of interest needs to be defined prior by a profound structural analysis. Consequently, experiments with quasi-realistic loads on structures of increased complexity (i.e., demonstrators) that reflect real components enable (i) cost-efficient studies of, e.g., damage parameters or environmental conditions, and (ii), a more reliable transfer of findings from simpler structures. Therefore, it is expected that this approach will allow robust and damage-sensitive SHM methods to be developed in a shorter time and in a more cost-effective manner. Other contributions [12] readily proposed the implementation of testing sensor systems and SHM methods in the structural test pyramid. However, the question of how to efficiently transfer knowledge from low-level tests to higher levels of complexity remains. The present work proposes a method that enables one to generate highly transferable test results at a comparatively low level of complexity and cost.

In the last decades, numerous SHM methods have been proposed and were successfully tested on simple specimens relevant to aircraft design in laboratory environments [13–18]. Recently, scaled demonstrators equipped with multiple sensors have also been built to develop and test the applicability of promising SHM methodologies [19–21]. SHM methods that are capable of monitoring large thin-walled structures, e.g., spoilers, are guided waves [22–26], electrical impedance tomography (EIT) through conductive surface layers [27,28], and direct measurements of a structure's electrical impedance [29]. Furthermore, strain-based methods with distributed strain sensors, e.g., fiber optical sensors (FOSs), are expected to efficiently monitor large thin-walled structures, which are typical for lightweight design [30–33].

This contribution presents the development of an idealized demonstrator on a 1:2 scale together with a concentrated load optimization methodology to mimic the deformation and strain states (with respect to strain directions) of a large civil aircraft spoiler under a specific aerodynamic load case. A similar algorithm was reported in [9] to define the loading for strength tests of a horizontal stabilizer, where the load amplitudes at a number of fixed introduction points were optimized to reflect a considered load case. However, the load optimization algorithm presented in this work enables one to find the best locations of a specific number of load introduction points. Furthermore, the presented methodology reduces high stresses in critical regions to allow high loads and, thus, high strain

values to increase the signal-to-noise ratio (SNR) in further investigations (if required). In addition to optimizing concentrated loads for demonstrators, a wide variety of optimization methodologies are used in aircraft design to optimize composite structures for weight reduction [34,35], the shape of aircraft parts to reduce gas leakage [36] and noise impact on the ground [37], or sensor positions for SHM applications [38,39], to name just a few examples.

The idealized spoiler demonstrator is needed to further develop and validate novel sensor systems and SHM methods to, e.g., detect and identify sandwich face-layer debondings and delaminations under realistic loading conditions [40,41]. Nowadays, spoilers of large civil aircraft are typically built as composite sandwich structures composed of glass-fiber-reinforced polymer (GFRP) face layers, a honeycomb core, and monolithic hinges for mounting and actuation [1]. Impact damages in monolithic composite structures caused by, e.g., dropping of a tool or a bird strike, as well as manufacturing defects, are usually difficult to detect through visual inspection. The same can be said for sandwich structures, where, in addition, debonding of the core and skin and deteriorations of the sandwich core are possible failure modes. A secure and common procedure for detecting structural damages that are not detectable through visual inspection in composite structures is through more sophisticated non-destructive testing (NDT) methods, e.g., ultrasonic testing, radiographic testing, vibration/modal analysis, etc. [42]. However, inspections of large areas using advanced NDT are labor expensive, and hence, they increase down time and maintenance costs. Using similar damage detection methods, SHM promises to overcome these issues by monitoring the aircraft structure continuously during operation with structurally integrated systems of sensors [18]. Furthermore, lightweight structures such as aircraft are highly optimized for specific loading scenarios. Hence, in the case of manufacturing defects, which most probably decrease structural strength, or in the case of overloads, such structures are especially vulnerable to failure.

The typical overall dimensions for spoilers of large civil aircraft depend on the specific aircraft type and location on the wing; e.g., Airbus A340 spoiler number 2 (highlighted in Figure 1) has dimensions of approximately 2400 mm × 800 mm × 150 mm.

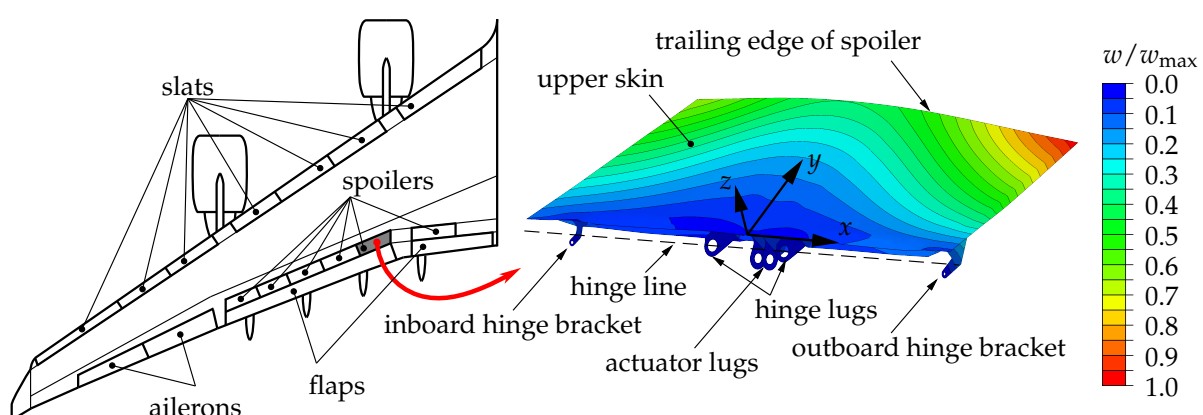

**Figure 1.** Location and deformation of the considered aircraft spoiler of an Airbus A340 aircraft, overview of control surfaces (cf. [43]), and out-of-plane deformation of the spoiler's upper skin according to considered load case.

Consequently, the acquisition of such spoilers and test rigs for mechanical loading is cost-intensive, and they are often not available in academic research, where SHM methods are currently developed. Furthermore, an adequate deformation and strain state (with respect to strain directions) can only be achieved with high loads, resulting in high potential strain energy, which can be a safety issue. A small-scale idealized demonstrator is expected to overcome these difficulties. Obvious advantages are the lower manufacturing costs, fast assembly, simple introduction of idealized artificial damages, easier handling, and smaller deformation forces due to reduced dimensions and a simplified geometry. However, such

demonstrators need to be tailored very well to the objectives of the scheduled investigation. This is of particular importance when the complexity of the considered structure, its loads, and the effects provoked by potential damage increase. Aerodynamic loads are always a challenge to realistically reproduce in mechanical tests. This is also true for the reconstruction of loaded states for SHM evaluations, as required by strain-based methods in general [30,31,44] and the zero-strain trajectory method [45–48] in particular. The experimental validation of the latter for the identification of sandwich-face-layer debonding and delamination is the long-term objective for the present work.

The reference considered for the development of a scaled demonstrator is a spoiler of an Airbus A340 wing. This specific control surface is a sandwich structure with a wedge-shaped honeycomb core. The upper skin and lower skin are fiber-reinforced polymer (FRP) lamina. The considered spoiler and an overview of the control surfaces of an Airbus A340 aircraft wing [43] are given in Figure 1. A design load case is considered as the reference load, where the spoiler has to withstand the aerodynamic pressure during landing. In this scenario, immediately after touchdown, the spoiler is extended 35° using a hydraulic cylinder attached to the actuator lugs. The given loading results in the out-of-plane displacement $w$ of the upper skin, which is depicted in Figure 1. The largest displacement $w_{\max}$ occurs at the out-board side of the trailing edge of the spoiler.

## 2. Materials and Methods

An idealized spoiler demonstrator should be developed to test SHM methods in laboratory experiments, which simulate spatial strain states comparable to strain states present on the upper skin of the real aircraft spoiler during landing. These strain states do not have to match in their absolute numbers, but the strain directions—i.e., strain trajectories along the, e.g., major principal strains—need to be reproduced. Furthermore, the test setup effort should be minimal to perform experiments at low costs and in a short time.

The real aircraft spoiler has a nonuniform cross-sectional shape (see Figures 1 and 2) and a heterogenous upper skin (thickness is not constant, multiple FRP lamina with different layups). Hence, the exact stress and strain amplitudes resulting from the nonuniform aerodynamic load can not be represented locally by, e.g., a simple sandwich panel with uniform thickness. However, to yield strain directions (and stress directions considering linear elastic material properties) similar to the real aircraft spoiler, it is sufficient for the idealized spoiler demonstrator to represent a similar deformation shape. More precisely, a similar out-of-plane deformation shape $w(x, y)$, i.e., for $w^S(x, y) = k\, w^D(x, y)$, where $k$ is a scaling factor, will result in a comparable curvature, e.g., $\partial^2 w(x, y)/\partial x^2$, of the sandwich plates (aircraft spoiler and idealized demonstrator; indicated by superscripts $S$ and $D$, respectively). Hence, a deformation, equally scaled on the whole spoiler surface, generates similar strain states—particularly with respect to the strain directions (amplitudes might deviate)—on the upper skins. This is true if all fractions ($\varepsilon_{xx}/\varepsilon_{yy}$, $\varepsilon_{xx}/\varepsilon_{xy}$, $\varepsilon_{yy}/\varepsilon_{xy}$ in the two-dimensional case) of two different strain tensors (aircraft spoiler and idealized demonstrator) are identical in every point of the upper skin, which is visualized by a simple scaling of Mohr's circle [45,46] and easily shown by opposing the bending strains of the plate elements ($S$ and $D$) in, e.g., the $x$ and $y$ directions.

$$\frac{\varepsilon_{xx}^S(x,y)}{\varepsilon_{yy}^S(x,y)} = \frac{-\frac{\partial^2 w^S(x,y)}{\partial x^2}\frac{h^S(x,y)}{2}}{-\frac{\partial^2 w^S(x,y)}{\partial y^2}\frac{h^S(x,y)}{2}} \overset{w^S(x,y)=k\,w^D(x,y)}{=} \frac{-\frac{\partial^2 w^D(x,y)}{\partial x^2}\frac{h^D(x,y)}{2}}{-\frac{\partial^2 w^D(x,y)}{\partial y^2}\frac{h^D(x,y)}{2}} = \frac{\varepsilon_{xx}^D(x,y)}{\varepsilon_{yy}^D(x,y)}, \tag{1}$$

where $h^S(x, y)$ and $h^D(x, y)$ are the thicknesses of the aircraft spoiler and the idealized demonstrator, respectively.

Beyond that, the equivalent stresses in all individual parts of the idealized spoiler demonstrator should stay within the elastic regime during loading to avoid the plastic deformation or even fracture of any component.

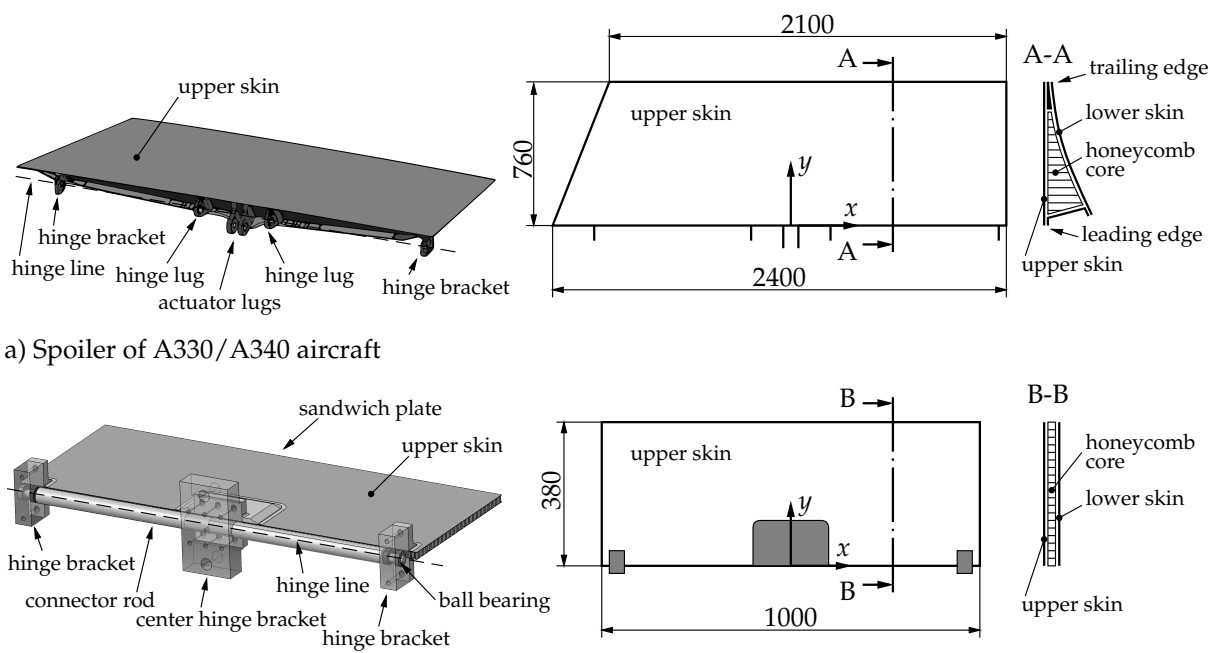

a) Spoiler of A330/A340 aircraft

b) Idealized demonstrator

**Figure 2.** Shape and dimensions of (**a**) the real aircraft spoiler considered in comparison to (**b**) the developed idealized spoiler demonstrator.

### 2.1. Structural Definition

In general, to establish a proper demonstrator, necessary simplifications must be well considered to fit the corresponding application. To realize a cost-efficient idealized demonstrator of the considered aircraft spoiler (shape and dimensions depicted in Figure 2a), which is feasible for applying novel SHM methods, three simplifications were made. First, the idealized spoiler demonstrator was scaled to fit a size smaller than $1\,\text{m} \times 1\,\text{m}$. A demonstrator of this size fits well into the available test rig and can be set up and operated by a single person. Second, a rectangular shape and symmetric loading is considered in order to obtain an efficient simulation model. Additionally, the chosen symmetry allows one to perform comparative measurements on both sides of the idealized spoiler demonstrator and reduce manufacturing costs. Third, the cross-section of the idealized spoiler demonstrator should be homogeneous to keep the manufacturing costs low. In this way, commercially available sandwich panels can be used for the idealized spoiler demonstrator. Therefore, the inhomogeneous triangular sandwich design of the real aircraft spoiler is replaced by the standard sandwich plate with a homogeneous cross-section, as depicted in Figure 2b. The center hinge bracket (CHB), which incorporates two hinge lugs and the actuator lugs, is replaced by two aluminum parts adhesively bonded to the upper and lower skin of the sandwich panel and a support block that connects these two parts. In addition, three aluminum blocks for each hinge bracket at the corners are used as additional supports for the idealized spoiler demonstrator. A connector rod (simple beam with circular cross-section) along the hinge line connects all supports. Although the CHB is rigidly mounted to the test rig, the two hinge brackets at both ends of the connector rod can rotate around the *x*-axis. The rotational degree of freedom is provided using ball bearings between support blocks and the connector rod.

### 2.2. Loading Definition and Optimization

The distributed aerodynamic load, which acts on the real aircraft spoiler, should be represented by a small number of concentrated loads (modeled on single nodes) at the idealized spoiler demonstrator to allow load application with a simple whiffle tree; see Section 3. Obviously, this simplification of loading (distributed load is replaced by

a small number of concentrated loads) is not feasible for representing the full structural behavior (internal forces, local strain, and stress amplitudes can be fundamentally different). However, in this case example, the reproduction of the out-of-plane deformation shape and, hence, the strain directions of the structure under investigation should be reproduced; see Equation (1). Based on a previous study, it is assumed that five concentrated loads should be sufficient to generate the desired deformation shape of the idealized spoiler demonstrator [49]. However, the exact optimal locations and amplitudes of these loads to reproduce the out-of-plane deformation shape of the real spoiler are unknown. Therefore, a multidimensional non-linear minimization with bound constraints by transformation was implemented in the numeric computing environment of Matlab® [50]. The algorithm was designed to identify the optimal locations and amplitudes of three concentrated loads acting on a symmetrical half model of the spoiler demonstrator based on parametric simulations by incorporating a simple finite element (FE) shell model that was solved in Abaqus/Standard [51].

### 2.2.1. Simple FE Shell Model

The search algorithm for the optimal load introduction uses a highly simplified FE model. Due to the symmetry in the $x$-plane and the $z$-plane of the idealized spoiler demonstrator, it is implemented as half model with planar shell elements; see Figure 3. The sandwich panel is modeled using a composite layup with an isotropic linear elastic material for the skin (thickness of 1 mm, $E_{Al} = 70$ GPa, $\nu_{Al} = 0.33$) and an orthotropic material definition for the core (thickness of 1 mm, $E_1 = 1$ MPa, $E_2 = 1$ MPa, $E_3 = 630$ MPa, $\nu_{12} = \nu_{13} = \nu_{23} = 0$, $G_{12} = 1$ MPa, $G_{13} = 280$ MPa, $G_{23} = 140$ MPa; see [52]). In the area of the hinge brackets (small black rectangles in Figure 3), the honeycomb core is replaced by an isotropic and linear elastic material model of steel ($E_{St} = 210$ GPa, $\nu_{St} = 0.3$). The boundary conditions of this simplified FE model are chosen to represent the boundaries of the real aircraft spoiler. All nodes of the CHB (black area in the center) are fixed in all degrees of freedom (DOFs). To allow a rotation and axial translation of the small hinge bracket at the corner, the nodes of its back edge are not restrained in the DOFs related to the $x$-direction. The FE model is loaded with one single concentrated unit force in the negative $z$-direction. This load is located at any single node in one of the highlighted regions, $\mathcal{J}, \mathcal{K}$, and $\mathcal{L}$, in Figure 3. The out-of-plane deformation ($z$-direction) of the shell model is simulated for every node position of the concentrated unit load.

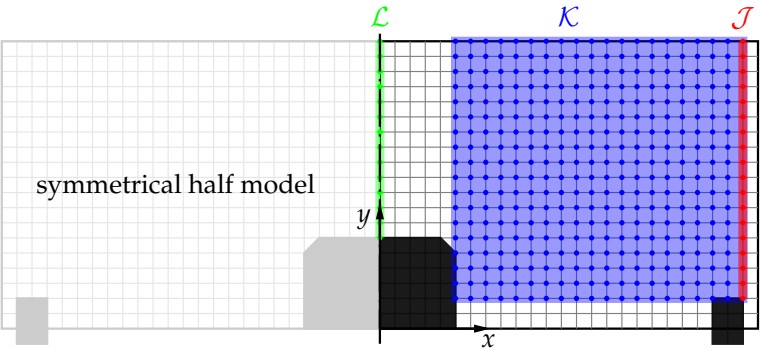

**Figure 3.** FE shell model with three predefined node sets, $\mathcal{J}, \mathcal{K}$, and $\mathcal{L}$. The 2D shell elements have an exact size of 20 mm × 20 mm.

### 2.2.2. Optimal Locations and Amplitudes of Concentrated Loads

In order to find loads that result in a deformation and, thus, spatial strain orientation, similarly to the real aircraft spoiler, the developed simple FE shell model is integrated into a minimization algorithm implemented in Matlab®. Herein, the out-of-plane deformation in a defined operating condition of the real aircraft spoiler acts as the target function; see Figure 1. The main objective is to minimize the difference between the numerically calculated out-of-plane deformation of the real aircraft spoiler and that of the spoiler

demonstrator FE shell model. The structure of the optimization algorithm is depicted in Figure 4.

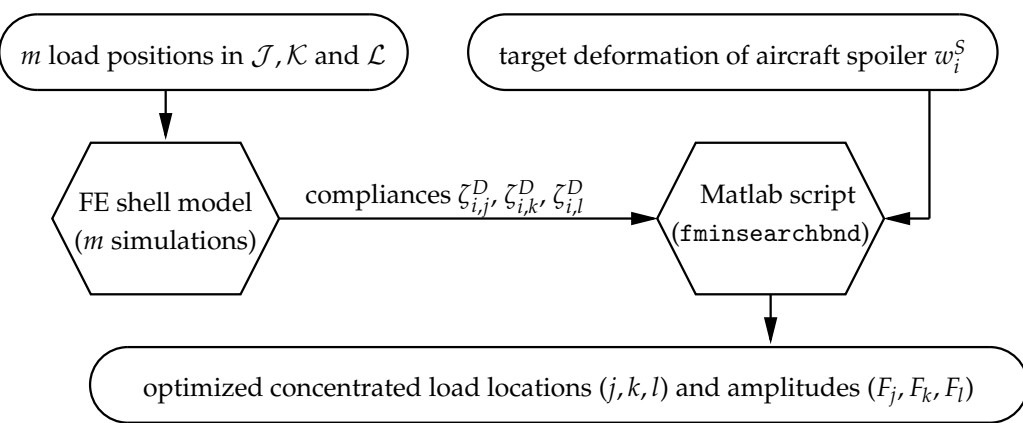

**Figure 4.** Flowchart of the proposed concentrated load optimization methodology.

First, the unit load is applied step by step to one node at a time for all nodes in $\mathcal{J}, \mathcal{K}$, and $\mathcal{L}$; see Figure 3. After the calculation of all FE shell models (with applied unit loads on $m = |\mathcal{J}| + |\mathcal{K}| + |\mathcal{L}|$ different locations), the resulting deformation results are exported for further analysis. Second, the parameter optimization

$$\underset{\{j\in\mathcal{J},k\in\mathcal{K},l\in\mathcal{L},F_j,F_k,F_l\}}{\arg\min}\left\{(F_j+F_k+F_l)\underset{\{0\leq F_j,F_k,F_l\leq 5000\}}{\min}\left[\sum_{i=1}^{n}\left(F_j\zeta_{i,j}^D+F_k\zeta_{i,k}^D+F_l\zeta_{i,l}^D-w_i^S\right)^2\right]\right\}=\{j,k,l,F_j,F_k,F_l\} \quad (2)$$

is executed using the Matlab® function `fminsearchbnd` [53], where $\zeta_{i,j}^D$, $\zeta_{i,k}^D$, and $\zeta_{i,l}^D$ are calculated compliances of node $i$ for each defined unit load at nodes $j \in \mathcal{J}$, $k \in \mathcal{K}$, and $l \in \mathcal{L}$. The inner minimization of Equation (2) represents a least-squares search with the superposition of compliances $\zeta_{i,j}^D$, $\zeta_{i,k}^D$, and $\zeta_{i,l}^D$ multiplied by unknown $F_j$, $F_k$, and $F_l$ (which yields the displacements of the demonstrator) and subtracted by the target displacements of the civil aircraft spoiler $w_i^S$ for all relevant nodes $n$. The additional constraint of $0 \leq F_j, F_k, F_l \leq 5000$ ensures forces in the negative $z$-direction (see Figure 5), as well as a limitation to a maximum load amplitude of 5000 N. Subsequently, the sum of the squared differences between the target deformations and the deformations of the demonstrator is weighted by the sum of the loads ($F_j$, $F_k$, and $F_l$) found. This energy-type expression is used to find an optimal solution that balances the deformation accuracy and required load sizes.

The resulting load amplitudes $F_j$, $F_k$, and $F_l$ and corresponding positions of nodes $j,k$, and $l$ on the upper skin of the idealized spoiler demonstrator are given in Table 1.

**Table 1.** Optimized concentrated loads and their locations on the idealized spoiler demonstrator according to Equation (2).

| Node | Load [N] | $x$ [mm] | $y$ [mm] |
|------|----------|----------|----------|
| $j$ | 1885 | 480 | 120 |
| $k$ | 2705 | 440 | 360 |
| $l$ | 0 | 0 | 120 |

As, at the symmetry line load $F_l$ yields a numerical value close to zero, the overall result is a four-point loading of the idealized spoiler demonstrator, which is depicted in Figure 5b. However, with the relatively large loads given in Table 1, the stress in the sandwich skin around the corner of the CHB exceeds the yield stress of the initially considered aluminum alloy ($R_{p02,\text{Al}} = 130\,\text{MPa}$). A proportional reduction of load amplitudes would

lead to a decrease in deformation and straining of the upper skin of the idealized spoiler demonstrator. In this case, strains in large areas of the upper skin calculated with the simple FE shell model would then be below 20 µm/m, which is defined as the minimum strain amplitude measurable with the facilitated digital image correlation (DIC) system (cf. Section 3). Furthermore, considerable strain amplitudes are desired in order to yield a large SNR for newly developed sensor systems.

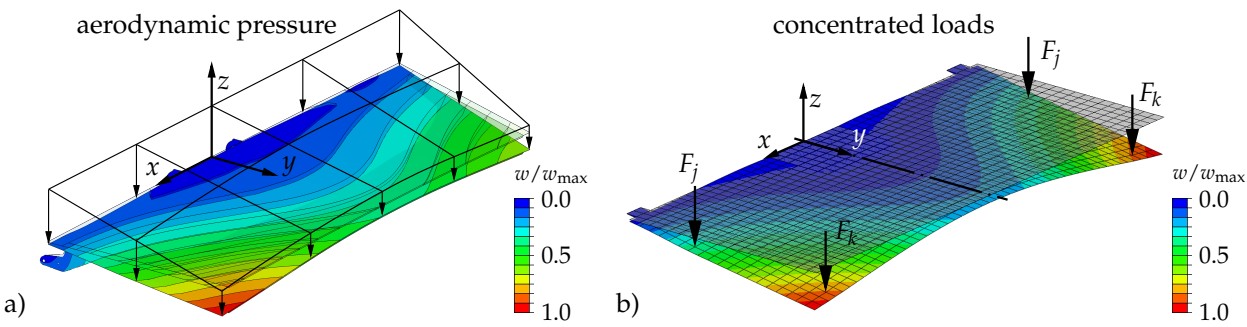

**Figure 5.** Result of loading optimization on an idealized spoiler demonstrator. Schematic sketch of (**a**) the real aircraft spoiler due to aerodynamic loads and (**b**) the idealized spoiler demonstrator under four-point loading (half model rendered in symmetrical full view for display purposes).

Thus, the sandwich panel made out of an aluminum alloy was replaced by a composite sandwich panel with an aramid honeycomb core and GFRP skins with a quasi-isotropic layup [0,45,−45,0] and a total skin thickness of 0.5 mm. The thickness reduction of the sandwich skin from 1.0 mm to 0.5 mm reduces the bending stiffness significantly and, thus, allows larger deformation at the same loading. The material parameters of the composite sandwich panel are given in Table 2.

**Table 2.** Material parameters of the composite sandwich panel of the idealized spoiler demonstrator [54].

| | $E_{11}$ [MPa] | $E_{22}$ [MPa] | $E_{33}$ [MPa] | $\nu_{12}$ - | $\nu_{13}$ - | $\nu_{23}$ - | $G_{12}$ [MPa] | $G_{13}$ [MPa] | $G_{23}$ [MPa] |
|---|---|---|---|---|---|---|---|---|---|
| Each layer of skin ([0,45,−45,0]) | 22,550 | 20,900 | 1 | 0.15 | 0 | 0 | 4500 | 3500 | 3500 |
| Core | 1 | 1 | 500 | 0 | 0 | 0 | 1 | 66 | 34 |

The change in the material brings two additional advantages: First, the given GFRP sandwich skins have an maximum allowable in-plane stress of $\sigma_{max,GFRP} = 100$ MPa, which is similar to the yield stress of the initially considered aluminum sandwich skin, thus allowing loading at similar stresses. Second, due to the smaller stiffness (more than three times) of the composite sandwich panel (cf. Table 2) compared to the stiffness of the aluminum alloy, a larger deformation and, hence, approximately 2.5 times larger strain amplitudes (considering stiffness and maximum allowable in-plane stresses) can be achieved with the same loading. However, the maximum possible strain amplitudes are defined by a further detailed analysis of strain states and stresses in each component using a three-dimensional (3D) FE model.

### 2.3. Stress and Strain Analysis with a Detailed 3D FE Model

The idealized four-point loading found with an optimization with a simplified FE shell model (cf. Section 2.2.1) is now applied to a more sophisticated symmetrical 3D FE model. Two concentrated loads are defined at the identified optimal locations (see Table 1) on the lower skin of the idealized spoiler demonstrator. By this measure, the local influence of concentrated loads on the deformation and strain states of the upper skin around the load introduction points (single nodes on lower skin) is minimized. The 3D FE model incorporates the updated composite sandwich geometry and material properties (cf. Table 2)

and includes many more details than the simple shell model representation used for the optimization of the loading. The geometry of the detailed half model of the spoiler demonstrator is depicted in Figure 6.

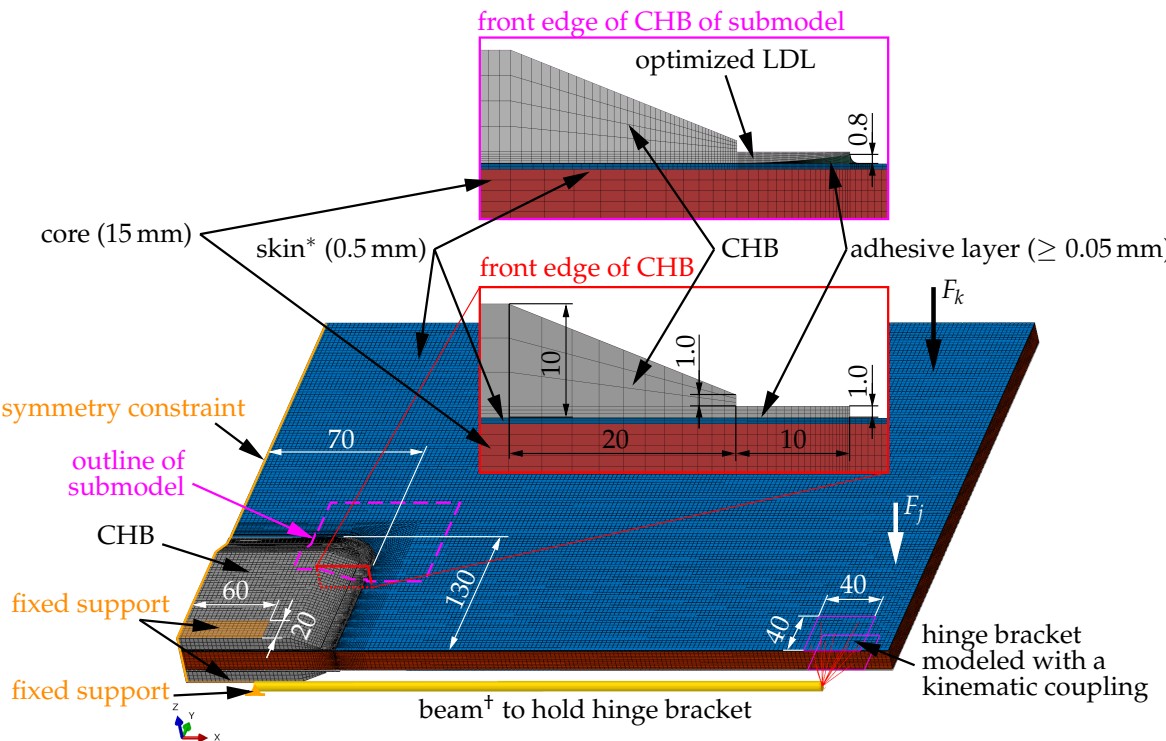

**Figure 6.** Detailed 3D FE model (all dimensions in millimeters), * shell thickness, and † beam cross-section rendered for display purposes.

This incorporates multiple parts: The sandwich panel is modeled with a solid core (8-node linear brick elements with reduced integration, C3D8R in nomenclature of the used FE program Abaqus [51], painted red in Figure 6), and the shell elements as top and lower skins (4-node linear shell elements with reduced integration, S4R in Abaqus nomenclature, painted blue in Figure 6) connected by tie constraints. Below each support block of the CHB (8-node linear brick elements with reduced integration, painted in gray in Figure 6) lies an adhesive layer with a thickness of 0.05 mm (8-node linear brick elements with reduced integration, painted green in Figure 6). Subsequently, to connect the support blocks of the CHB (painted gray in Figure 6) with the sandwich skins, tie constraints between the skin and adhesive layer and between the adhesive layer and support block are used. The connector rod, which holds the hinge bracket of the idealized spoiler demonstrator, is modeled by a beam with constant circular cross-section (diameter $d = 35$ mm and length $l = 400$ mm; 2-node linear beam elements, B31 in Abaqus nomenclature, painted yellow in Figure 6), as well as a linear elastic material model of steel ($E_{St} = 210$ GPa, $\nu_{St} = 0.3$). The support blocks for the CHB and the hinge brackets are modeled with linear elastic material behavior and parameters for aluminum ($E_{Al} = 70$ GPa, $\nu_{Al} = 0.33$). The parameters for the linear elastic material model of the adhesive layer (3M DP490 Epoxy, $E_{Ad} = 660$ MPa, $\nu_{Ad} = 0.38$) are taken from [55]. The boundary conditions for the detailed 3D FE model are highlighted in orange in Figure 6. Both CHBs are fixed in all DOFs at nodes in rectangular areas of size $20 \times 60$ mm$^2$, which represents the cross-section of the remaining CHB (cf. Figure 2b). A symmetry constraint is defined for all nodes of all components touching the $x$-plane. The left end of the connector rod is fixed in all DOFs. The nodes of the upper and lower skins of the sandwich panel, which lie in the area of the hinge bracket, are connected with a kinematic coupling and can only rotate around the bearing point at the right end of the connector rod; see Figure 6.

For the present idealized spoiler demonstrator, the highest stress amplitudes are calculated in the composite sandwich skin and the adhesive layer in a small region below the rounded corner of the CHB. A load distribution lip (LDL) with a cross-section of $10 \times 1\,\text{mm}^2$ is added to the front edge of the CHB in order to reduce these local high stress amplitudes. Additionally, an FE submodel of the corner region (node-based submodeling technique with an enforced displacement boundary condition; outline indicated by a dashed magenta line in Figure 6) was used to improve the shape of the LDL. With this detailed FE submodel, different LDL types were tested (various thicknesses of the LDL, different fillet and chamfer types—not depicted in Figure 6). The best shape found has a large chamfer, which tangentially reduces the thickness of the LDL from 1.0 to 0.25 mm and increases the thickness of the adhesive layer from 0.05 to 0.8 mm (cross-section of the improved LDL and adapted adhesive layer at the front edge of the CHB are depicted in the top of Figure 6). This improved LDL shape reduces the local high stresses in the adhesive layer by 33 % and in the upper skin by 28 % compared to the initial straight shape. With these two measures (change from aluminum to composite sandwich panel and adding an improved LDL), the maximum possible loads are $F_{j,\text{max}} = 178.6\,\text{N}$ and $F_{k,\text{max}} = 256.4\,\text{N}$.

The out-of-plane deformations of the upper skin calculated with the detailed 3D FE model of the idealized spoiler demonstrator and a comparison with deformations given for the FE model of the real A340 aircraft spoiler are presented in Figure 7. For better comparability, both contour plots are normalized to the out-of-plane displacement calculated for the same point $P_0 = (x = 500\,\text{mm}, y = 380\,\text{mm})$. The white dashed rectangles indicate the outer dimensions of the idealized spoiler demonstrator. A close correlation between both deformation contours is shown. The out-of-plane deformation in point $P_0$ calculated for the idealized spoiler demonstrator at maximum loading yields $w_{\text{FEM}}(P_0) = 22.93\,\text{mm}$.

Real aircraft spoiler (mirrored, scale 1:2)    Idealized spoiler demonstrator

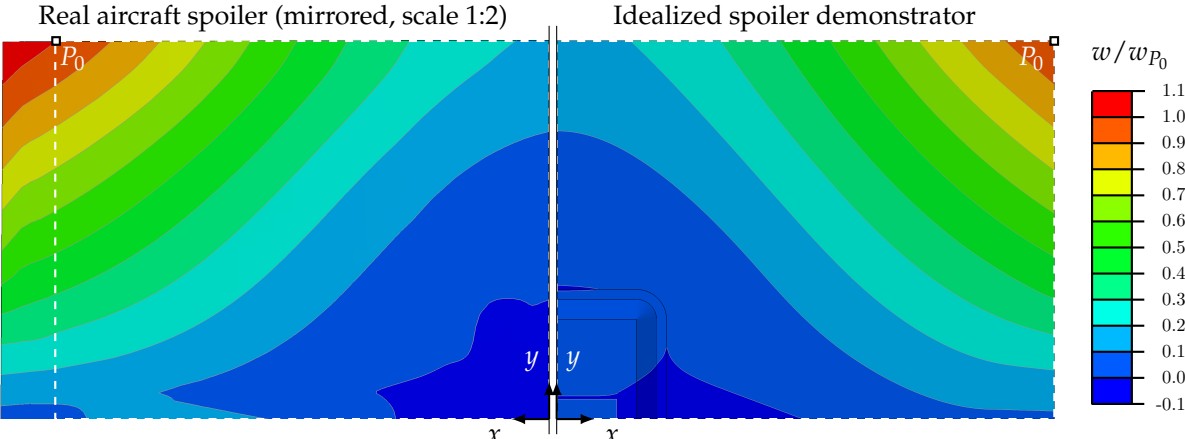

**Figure 7.** Out-of-plane displacement contour plots of numerical FE models (dashed rectangles indicate the size of the idealized spoiler demonstrator).

The evaluation of strain states is performed by using strain directions and strain trajectories. Aside from commonly known principal strain directions with angles $\alpha_{1,2}$ resulting from

$$\varepsilon_{nm} = -\frac{\varepsilon_{xx} - \varepsilon_{yy}}{2} \sin 2\alpha + \varepsilon_{xy} \cos 2\alpha \overset{!}{=} 0, \tag{3}$$

the so-called zero-strain directions $\beta_{A,B}$ are calculated with

$$\varepsilon_{nn} = \frac{\varepsilon_{xx} + \varepsilon_{yy}}{2} + \frac{\varepsilon_{xx} - \varepsilon_{yy}}{2} \cos 2\beta + \varepsilon_{xy} \sin 2\beta \overset{!}{=} 0, \tag{4}$$

where $\varepsilon_{xx}$, $\varepsilon_{yy}$, and $\varepsilon_{xy}$ are components of the linear strain tensor. The right-hand sides of Equations (3) and (4), i.e., $\overset{!}{=} 0$, indicate that the solutions for angles $\alpha$ and $\beta$ are found when $\varepsilon_{nm}$ and $\varepsilon_{nn}$ are set to zero, respectively. The normal strain vanishes in both zero-strain

directions $\beta_A$ and $\beta_B$. Note that, in the zero-strain directions, the second normal strain component and the shear strain component of the calculated strain tensor are not zero. Strain trajectories start at an arbitrary point and are calculated by following a specific strain direction in an iterative manner [45–47].

A comparison of the numerically calculated strain directions and the three selected trajectories of the aircraft spoiler with aerodynamic pressure loading and the idealized spoiler demonstrator with four-point loading is depicted in Figure 8. In front of the CHB (in the center of the spoiler surface), no zero-strain directions can be computed because the strains in both principal directions have positive signs. Therefore, instead of zero-strain directions, the minor principal strain directions are used in this area. This area in front of the CHB, where no zero-strain trajectories exist, is larger for the idealized spoiler demonstrator than for the real aircraft spoiler. In addition, in the vicinity of the free edges of the real spoiler surface, the strain directions yield larger deviations than on most of the spoiler surface around the CHB. However, the overall shapes of trajectories fit well together. The transitions between zero-strain and principal strain trajectories are also located at similar locations on the spoiler surfaces.

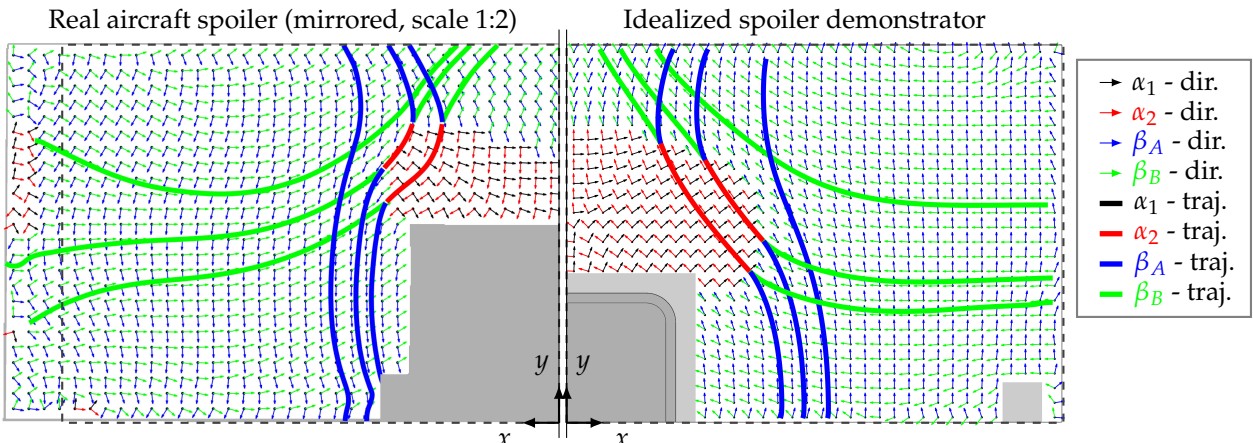

**Figure 8.** Comparison of strain directions and trajectories of numerical FE models (the region around the CHB and the hinge bracket is not considered; dashed rectangles indicate the size of the idealized spoiler demonstrator).

## 3. Experimental Validation of the Developed Idealized Spoiler Demonstrator

The experimental validation of the numerical results of the idealized spoiler demonstrator is performed by means of mechanical tests and a DIC system. In this section, first, the assembly steps for building the idealized spoiler demonstrator are explained. Second, a detailed description of the experimental setup, including the demonstrator and measurement equipment, is given. Finally, the last paragraph of this section explains the measurement procedure performed.

### 3.1. Assembly of the Idealized Spoiler Demonstrator

The assembly of the idealized spoiler demonstrator is depicted in Figures 2b and 9. The aluminum supports (CHB and hinge brackets) were bonded onto the sandwich panel using the two-component epoxy adhesive 3M Scotch-Weld DP490 [56]. According to the data sheet, at least seven days of curing time at room temperature were given to achieve full strength of the bonding layer. After the curing period, filets with $R = 1\,\text{mm}$ at the edge around the support blocks were carefully machined using a ball-nose cutter to produce a clean and defined border. All support blocks ($2 \times \text{CHB}$ and $4 \times \text{hinge brackets}$) were mounted with M12 screws to aluminum blocks, which were themselves combined by the connector rod; see Figure 2b. The connector rod was rigidly mounted to the aluminum block of the CHB. Both support blocks for the hinge brackets were mounted with ball bearings onto the connector rod.

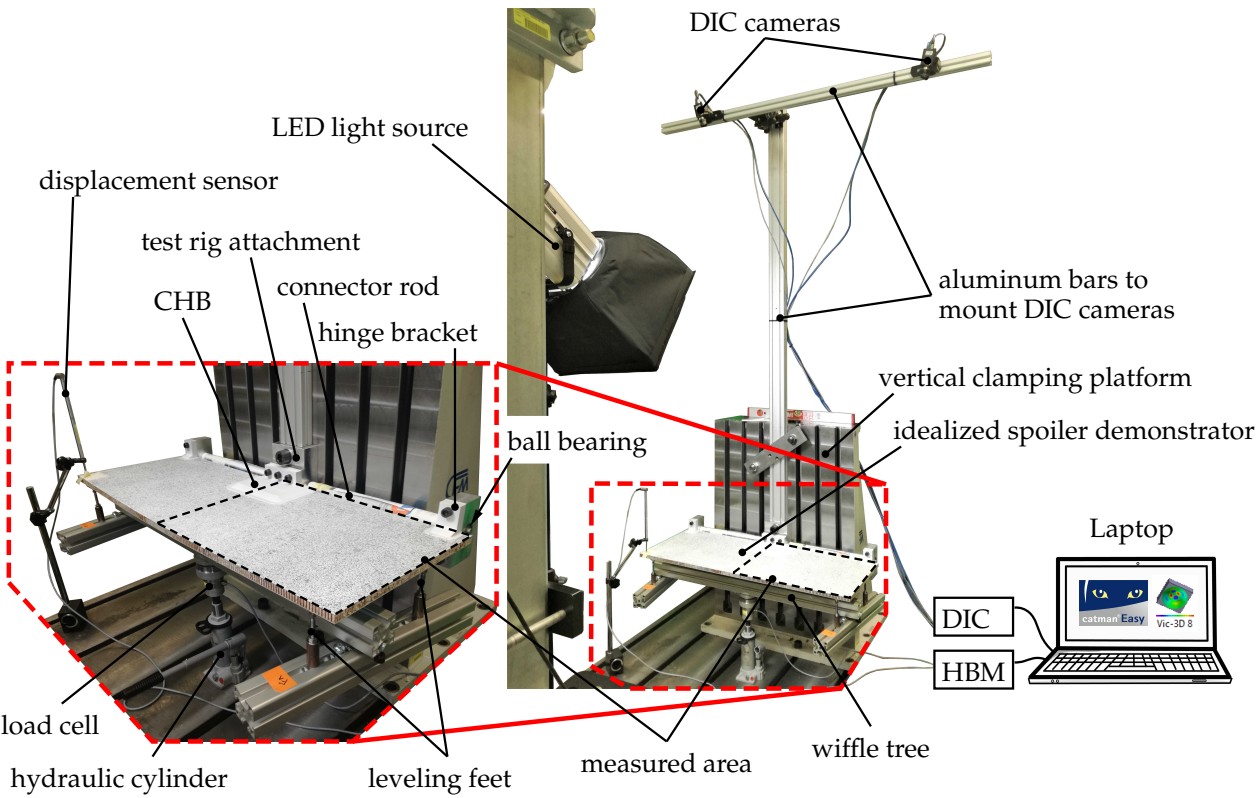

**Figure 9.** Test setup for the deformation and strain measurements.

### 3.2. Experimental Setup

The test setup, including the spoiler demonstrator and all facilitated measurement devices, is displayed in Figure 9. The complete assembly of the idealized spoiler demonstrator was mounted with two M24 screws and slot nuts to the vertical clamping platform of the test rig. The loading of the spoiler demonstrator was done with a single manual hydraulic cylinder. The load was distributed to the defined locations with a whiffle tree, as readily mentioned. Its arms were adjusted according to the calculated loads and their locations in Table 1. Each of the four loads was applied torque-free to the lower skin of the idealized spoiler demonstrator using steel leveling feet with a base plate diameter of 38 mm.

The facilitated measurement equipment included a load cell (HBM U3: $F_{max} = 10\,kN$) and a displacement sensor (HBM WA50: $d_{max} = 50\,mm$), both connected to the data acquisition device (HBM QuantumX MX840A: sample rate 100 Hz; cf. [57]). The load cell was located between the hydraulic cylinder and the whiffle tree. The displacement sensor was positioned at location $P_1 = (x = 480\,mm, y = 360\,mm$; compare Figures 9 and 10. Measurements of HBM sensors were recorded using the software HBM catman®Easy, cf. [58]. Additionally, a DIC system from Correlated Solutions Inc. with two synchronized cameras with a resolution of $2448 \times 2048$ pixels was used together with a HEDLER Profilux LED1000 light source [59]. The evaluation software VIC-3D 8 was used to analyze the DIC pictures that were taken. A speckle pattern was applied with an airbrush on the sandwich's upper surface using white as a background color and black as the speckle color (mean speckle size was 2.2 px). The cameras were mounted on the horizontal aluminum bar with a distance from each other of $l_{cam} = 995\,mm$ and a stereo angle of $\beta_{cam} = 27.9°$. The aluminum bar was adjusted parallel to the spoiler surface with a normal distance of $l_\perp = 2000\,mm$. With these camera positions, the complete left half of the spoiler surface could be measured by the DIC system (the area of measurement is indicated with a black dashed polygon in Figure 9).

Before starting the measurement, the whiffle tree was adjusted according to the calculated positions of the single load points, and all sensor signals were zeroed. Subsequently, the data acquisition was started, and the loading was manually increased until a total load of $F = 400\,\text{N}$ ($F_j = 82\,\text{N}$, $F_k = 118\,\text{N}$) was reached.

## 4. Results and Discussion

The results of measured displacements and strains are compared with the simulation results of the detailed 3D FE model. A comparison of the simulation results of the real aircraft spoiler and the 3D FE model has already been presented in Section 2.3, and will not be repeated here.

### 4.1. Out-of-Plane Displacements

A comparison of the displacements at a loading of $F = 2F_j + 2F_k = 400\,\text{N}$ calculated with the FEM simulation and measured with the DIC system is depicted in Figure 10. The overall shapes of the contour plots of the simulation and experiment show a very good match. The extraction of the displacement at point $P_2 = (x = -480\,\text{mm}, y = 360\,\text{mm})$ from the DIC measurement yields $w_{\text{DIC}}(P_2) = 11.19\,\text{mm}$. At the equivalent location on the opposite side of the idealized spoiler demonstrator (where no DIC measurements were conducted), the displacement sensor at position $P_1 = (x = +480\,\text{mm}, y = 360\,\text{mm})$ measured a similar displacement of $w_{\text{DS}}(P_1) = 11.14\,\text{mm}$. The deviation between the measured displacements at points $P_1$ and $P_2$ was less than $0.5\,\%$. Hence, these measurements at symmetrical points, as well as the similar displacement shapes, indicated that the whiffle tree was correctly adjusted and positioned.

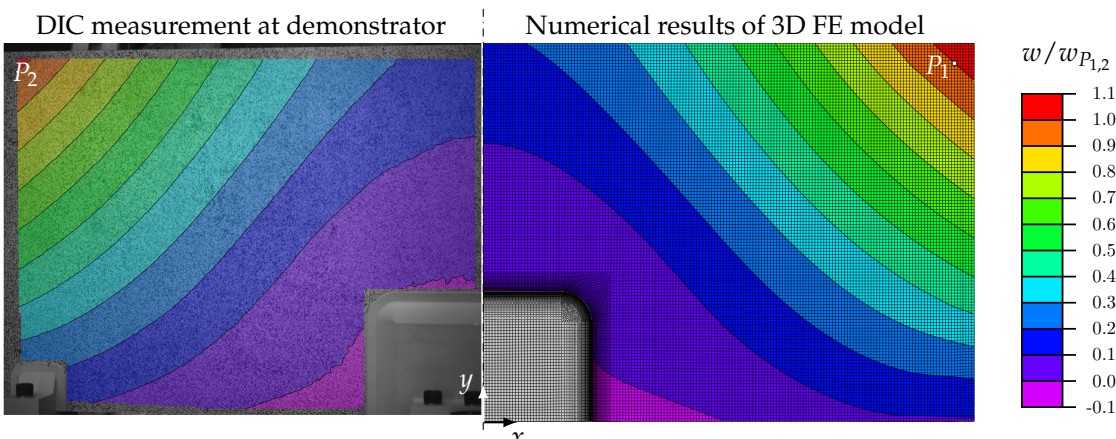

**Figure 10.** Out-of-plane displacement contour plots at an applied load of $F = 400\,\text{N}$.

The displacement calculated with the symmetrical 3D FE half model yielded $w_{\text{FEM}}(P_{1,2}) = 9.66\,\text{mm}$. The maximum deviation between the calculated and measured displacements in the simulation and experiments was $13.7\,\%$. This rather large deviation was expected because the material parameters were taken directly from data sheets and were not extracted from coupon tests. With a simple scaling of stiffness parameters, the deviation in amplitudes could be significantly reduced. However, more important for the development of a spoiler demonstrator to test SHM systems is a similar displacement shape, which was achieved with the current idealized spoiler demonstrator and the optimized four-point loading. Nevertheless, the final aim was to correctly represent strain states on the spoiler surface in order to test strain-based SHM methods.

### 4.2. Principal In-Plane Strains

Principal in-plane strains on the surface of the upper skin of the idealized spoiler demonstrator are depicted in Figure 11 for the DIC measurements and numerical results of the 3D FE model.

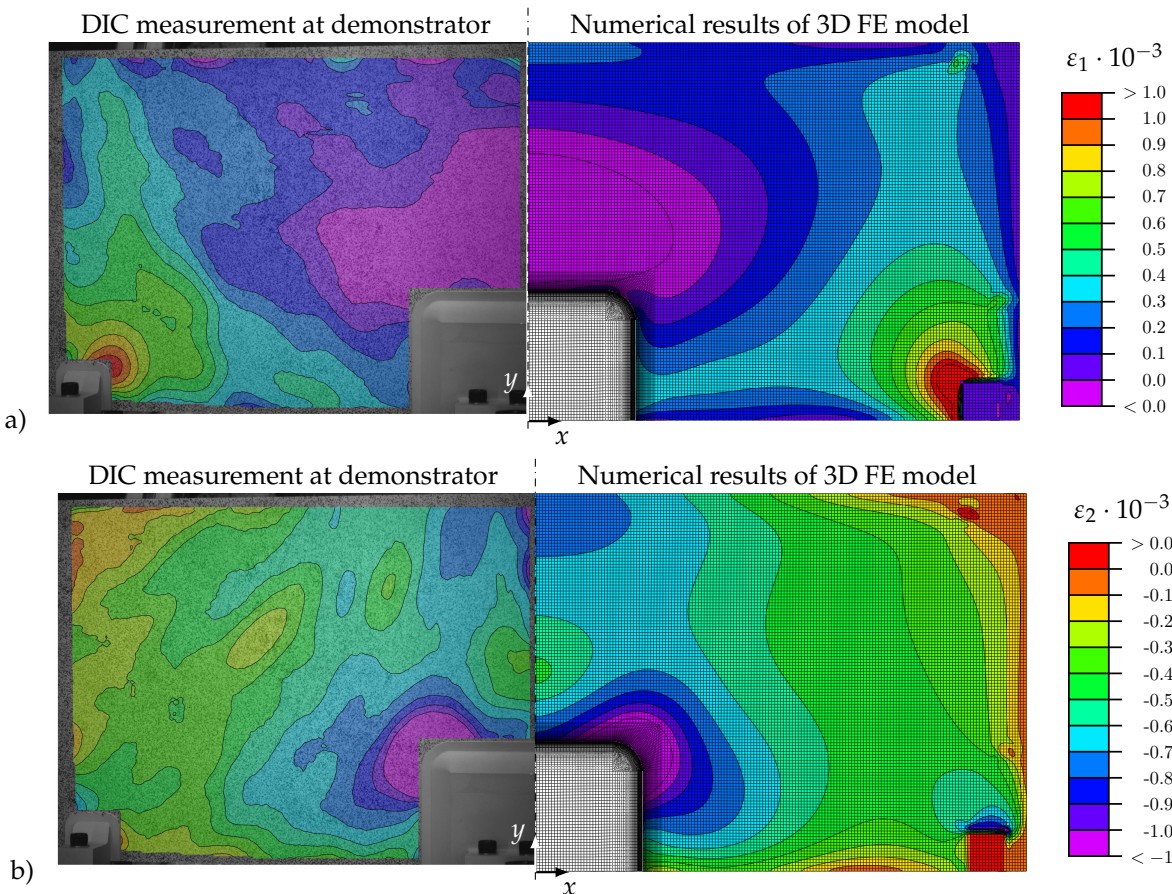

**Figure 11.** Comparison between the experiment and simulation of (**a**) minor and (**b**) major principal in-plane strains at an applied load of $F = 400\,\mathrm{N}$.

In order to measure the depicted area of the idealized spoiler demonstrator, the DIC cameras had to be positioned with a relatively large normal distance (see Section 3), which reduced the strain accuracy and resolution. Therefore, the calculated strain contours of the finite element simulation yielded smoother strain distributions and showed the load introduction points more clearly than the strain contours measured with the DIC system. However, in large areas, the major and minor principal in-plane strains showed similar results. The major principal strains, which are depicted in Figure 11a, mainly yielded positive amplitudes in most areas, except for the area in front of the CHB. In contrast, the minor principal strains exclusively yielded negative results on the whole surface; see Figure 11b.

The calculated and measured strain directions and resulting trajectories are depicted in Figure 12, showing the mirrored results of the FEM simulations for better comparison (otherwise, the notation of strain directions $\beta_A$ and $\beta_B$ would be interchanged; cf. Figure 8). In the center of the spoiler in front of the CHB, no zero-strain directions could be computed because the strains in both principal directions had negative signs; compare with Figure 11. On the remaining spoiler surface, only zero-strain directions are printed for better plot clearness. All depicted strain directions yielded similar orientations for the DIC measurement and the numerical simulation results. Furthermore, in the simulation and experiment, almost identical trajectories could be drawn along the zero-strain and major principal strain directions. It is assumed that distributed strain sensors, e.g., FOSs, applied along such lines can be used to efficiently monitor the structural integrity of the spoiler.

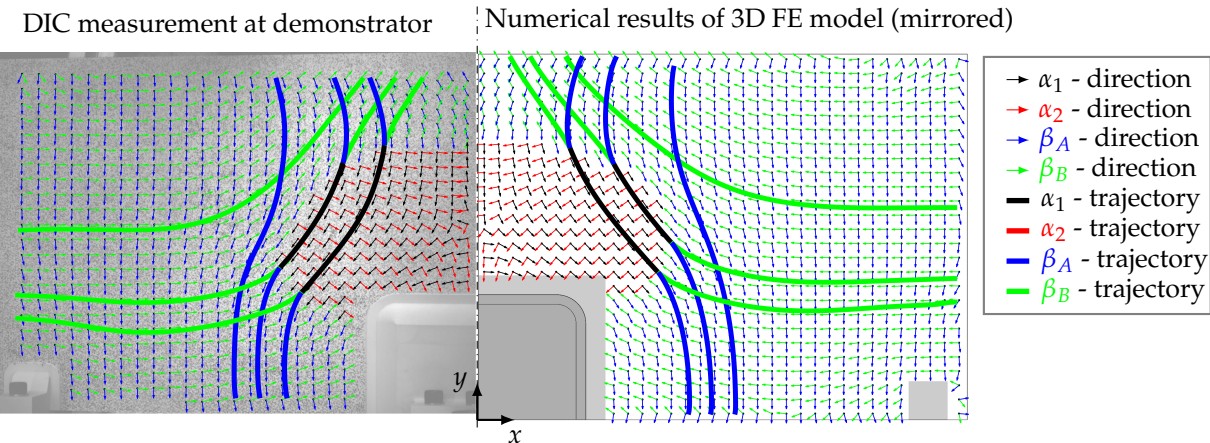

**Figure 12.** Comparison of the strain directions and trajectories for the experiment and simulation at an applied load of $F = 400\,\text{N}$.

## 5. Conclusions

The calculated deformation shape of a real aircraft spoiler subjected to aerodynamic pressure could be reproduced by means of a simplified and idealized spoiler demonstrator (homogeneous sandwich plate with attachments on a scale of 1:2) and four concentrated loads defined by location and amplitudes. This was achieved with a numerical FE-parameter-study-based optimization of the amplitudes and locations of five initially defined unit loads in order to minimize the out-of-plane deformation differences between a real aircraft spoiler and an idealized spoiler demonstrator by following a least-squares approach. Furthermore, the optimal load positions and amplitudes were weighted with the associated maximum stresses to enable large spatial strains without the structural failure of the demonstrator. Thus, deformations resulting from the four-point loading of the idealized spoiler demonstrator that were large enough to allow a distributed strain analysis of DIC measurements in mechanical tests were achieved. However, to further maximize the possible strain values, a composite sandwich with GFRP face layers and an aramid honeycomb core was used. Thorough stress and strain analyses were performed using a detailed 3D FE model. In a static strength analysis, special care was also given to the stresses in the adhesive layer between the sandwich panel and the CHB in the region around the front corner, where stress concentrations occurred. Through the design of a load distribution lip, the maximum stress in the adhesive between the CHB and sandwich face layer was reduced by 33 %. A comparison between the numerical results of the aircraft spoiler and the idealized spoiler demonstrator showed good agreement of the out-of-plane displacements and strain states with respect to strain orientations. Hence, for the aerodynamic pressure load considered, the displacements and spatial strain orientations of the real aircraft spoiler were adequately reconstructed with only four concentrated loads in the idealized spoiler demonstrator.

The validation of the simulation results for the idealized spoiler demonstrator developed here was performed by comparing the numerically calculated deformations and strain states with measurements gained from a corresponding experimental setup. The out-of-plane displacements were measured using a displacement sensor and a DIC system, and they fit well to the calculated results. Further processing of the measured strain states revealed that the strain directions and trajectories also correlated well with the FE results. However, the presented load optimization algorithm is also capable of quasi-realistically reproducing other loading conditions and strain states of operated aircraft spoilers. Hence, the idealized spoiler demonstrator and the load optimization algorithm represent a cost-efficient and adequate platform for experimental studies of SHM systems under quasi-realistic loading conditions and strain states.

The next step of this research work is the application of various sensors and SHM methods to the idealized spoiler demonstrator. In order to identify damages critical to

composite structures (e.g., sandwich debonding, delamination, impact damage), strain sensors must be applied to the idealized spoiler demonstrator, and potential strain-based SHM methods must be tailored to the considered structures and damage cases; they must also be applied and tested under changing environmental conditions. Finally, the most promising SHM methods that are the most cost-efficiently developed or have been improved using the spoiler demonstrator can be validated through applications on a real aircraft spoiler.

**Author Contributions:** Conceptualization, M.W., C.K., and M.S.; methodology, M.W. and C.K.; validation, M.W. and C.K.; investigation, M.W. and C.K.; writing—original draft preparation, M.W.; writing—review and editing, M.W., C.K., and M.S.; visualization, M.W.; supervision, C.K. and M.S.; funding acquisition, M.S. All authors have read and agreed to the published version of the manuscript.

**Funding:** This research was funded by the Christian Doppler Research Association, the Austrian Federal Ministry for Digital and Economic Affairs, and the National Foundation for Research, Technology, and Development.

**Institutional Review Board Statement:** Not applicable.

**Informed Consent Statement:** Not applicable.

**Data Availability Statement:** The data presented in this study are available upon request from the corresponding author.

**Acknowledgments:** The authors thank Erich Humer and Reinhold Wartecker for their precise manufacturing of the idealized spoiler demonstrator for the experimental tests. Furthermore, the support of Lukas Heinzlmeier in performing the experimental measurements, as well as that of Thomas Bergmayr and Martin Meindlhumer in reviewing the manuscript, is gracefully acknowledged. The authors thank Open Access Funding by the University of Linz.

**Conflicts of Interest:** The authors declare no conflict of interest.

## Abbreviations

The following abbreviations are used in this manuscript:

| | |
|---|---|
| SHM | structural health monitoring |
| FEM | finite element method |
| EIT | electrical impedance tomography |
| FOS | fiber optical sensor |
| FRP | fiber-reinforced polymer |
| GFRP | glass-fiber-reinforced polymer |
| NDT | non-destructive testing |
| SNR | signal-to-noise ratio |
| FE | finite element |
| DOF | degree of freedom |
| CHB | center hinge bracket |
| DS | displacement sensor |
| DIC | digital image correlation |
| 3D | three dimensional |
| LDL | load distribution lip |
| Al | aluminum alloy |
| St | steel |
| Ad | adhesive |
| cam | camera of the DIC system |

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
