# Peer review of "Development of Aircraft Spoiler Demonstrators for Cost-Efficient Investigations of SHM Technologies under Quasi-Realistic Loading Conditions"

_aerospace, doi:10.3390/aerospace8110320_

Round 1
Reviewer 1 Report
The paper aimed to closely reproduce the deformation shape of a real aerodynamically loaded aircraft spoiler by using an idealized 1/2 scale demonstrator with only four concentrated loads and a numerical parameter optimization algorithm. First, the lab-scale demonstrator was developed and simulated to reflect strain states of the real structure at operational loading conditions. Then, the experimental validation was performed and the measured deformations and spatial strain orientations were compared with the finite element simulations. The good correlation between numerical and experimental results showed a cost-efficient way to apply and test various sensor systems that are developed for structural health monitoring (SHM).
Overall, the manuscript is organized, well-written, and -presented. The study tries to develop a method for cost-efficient experimental studies on full-scale aircraft structure, yet the novelty of this method seems unclear in the current manuscript. Therefore, some revisions are needed before it is accepted for publication in the journal. The followings are the comments or issues existing in this version.
- The authors state that scaled demonstrators are commonly used to represent parts, assemblies, or full-scale structures to allow cost-efficient testing in aircraft engineering. The reviewer cannot clearly see the novelty of the proposed method. Some descriptions should be extended to strengthen the manuscript.
- Both the numerical and experimental models have been applied some simplification. How can these results be connected to a full-scale aircraft structure and finally achieved cost-efficient experimental studies for them?
- Please make the words or terminologies consistent, for example, “full scale” in Abstract and “full-scale” in Introduction.
- As a key of the cost-efficient idealized demonstrator, the authors should give more details about the four simplifications mentioned in Section 2.1. For example, the second simplification gives readers a quite comprehensive description.
- To understand the numerical parameter optimization algorithm proposed in the manuscript, a flow chart could be a great help for the readers.
- The symbol combined with an exclamation mark and an equal mark in Equation (2) and (3) should be defined.
- The authors are encouraged to add some paragraphs to describe the scope of Section 3 between line 318 (Section 3) and line 319 (Section 3.1).
- Last, there are few grammatical errors and typos in the current manuscript. For example, in line 347, “an HEDLER Profilux LED1000 light source” should be “a HEDLER Profilux LED1000 light source”, in line 394, “yield more smoother strain distributions” should be “yield smoother strain distributions”, and so on. Please revise them.
Reviewer 2 Report
- A pyramid figure representing the different scales from small coupons to large scale structure would be appreciated. there is plenty available in the literature.
- The introduction is very well written. a few more references on using optimization - data-driven - ai-based methods on SHM and composites can be added, as that explains the scaling work more.
- Lines 177 - Cite the source for the properties
- cite the source for properties in table 1
- citation references are required for - software used such as matlab, equipment used - such as line line 342, etc, and for materials such as in line 322, references to data sheets or websites of these products would be recommended.
- Line 380 - The error of 13.7% is large and not acceptable for aerospace applications. however proper reasoning has been given here. is future work also been planned to do on coupon tests ? or some data can be obtained from literature for coupon tests of samples and used for construction and validation of these models. - This is not a comment which you need to change in the article, just a suggestion from the reviewer.
